# Structural Aspects of Execution and Thermal Treatment of Welded Joints of Hardox Extreme Steel

**Łukasz Konat**

Department of Materials Science, Welding and Strength of Materials, Wrocław University of Science and Technology, 50-370 Wrocław, Poland; lukasz.konat@pwr.edu.pl

**Abstract:** The paper presents structure and mechanical properties of welded joints of the high-strength, abrasive-wear resistant steel Hardox Extreme. It was shown that, as a result of welding this steel, structures conducive to lowering its abrasion-wear resistance are created in the heat-affected zone. Width of the zone exceeds 60 mm, which results in accelerated wear in the planned applications. On the grounds of the carried-out examinations of structures and selected mechanical properties, a welding technology followed by heat treatment of heat-affected zones was suggested, leading to reconstruction of *HAZ* structures that is morphologically close to the base material structure. In spite of high carbon equivalent (*CEV*) of Hardox Extreme, the executed welding processes and heat treatment did not result in the appearance, in laboratory conditions, of welding imperfections in the welded joints.

**Keywords:** wear-resistant martensitic steel; submerged arc welding (SAW); heat treatment; structures; hardness changes; mechanical properties; Hardox Extreme steel

## 1. Introduction

During recent years, metallic materials classified by their manufacturers as low-alloyed abrasive-wear resistant martensitic steels, have been more and more widely applied. Regardless of their declared abrasion resistance, the common features of all these materials are very high mechanical parameters, which are maintained even for very thick steel sheet material. This feature is obtained with strictly selected chemical compositions depending on sheet thickness (in particular, the microaddition of 0.002% to 0.005% of boron), a reduced content of harmful admixtures of phosphorus and sulfur, and also by thermo-mechanical treatment. Results from generally available advertising information [1] and the author's own experience suggest that the currently available commercial steels of the considered material group reach a tensile strength exceeding 2000 MPa, with maintained satisfactory plasticity and impact strength. It is also worth stressing that these indices are obtained for carbon content up to 0.50 wt%, which is of crucial importance from the viewpoint of welding techniques. Tables 1 and 2 show properties of selected abrasive-wear resistant steels, which are declared by their manufacturers, and chemical compositions of these steels. In Table 2, in addition to the most commonly used *CEV* carbon equivalent (determining the metallurgical weldability of steel according to the International Institute of Welding), the carbon equivalent *CET* is also given. This indicator, according to the SS-EN 1011-2 standard, determines the preheating temperature to avoid hydrogen cracking of welded joints of fine-grained, non-alloyed, and low-alloy steels.

Information from manufacturers [1–5], literature data [6–9] and our own results [10–15] concerning the steels Hardox 400 and Hardox 500 confirm their good weldability and relatively high mechanical properties of welded joints. However, in each of the considered cases, thermal processes occurring during welding caused adverse structural changes in heat-affected zones of the welded materials, resulting in a significant reduction of their abrasive-wear resistance. Results of some research works [16]

suggest that changes of structure and hardness distribution occurring in welded joints of low-alloyed martensitic steels, adverse from the viewpoint of abrasive-wear resistance, can be eliminated by the application of additional heat-treatment operations. With regard to this, the author believes that it is worth complementing knowledge about execution and optimization of properties of welded joints of Hardox Extreme steels. In the previously published examination results of Hardox 600 [17], the authors indicated that it was possible to obtain static tensile strength of welded joints over 1500 MPa by proper selection of welding conditions and parameters, followed by suitable heat treatment.

**Table 1.** Declared mechanical properties of selected abrasive-wear resistant steels [1–5].

| Grade of Steel | $R_{p0.2}$ [MPa] | $R_m$ [MPa] | $A_5$ [%] | $KCV_{-40}$ [J/cm$^2$] | HBW |
|---|---|---|---|---|---|
| Hardox 400 | 1100 | 1250 | 10 | 56 | 370–430 |
| Hardox 450 | 1200 | 1400 | 10 | 50 | 425–475 |
| Hardox 500 | 1400 | 1550 | 10 | 46 | 470–530 |
| Hardox 600 | 1650 | 2000 | 7 | 25 | 570–640 |
| Hardox Extreme | NA | NA | NA | NA | 650–700 |
| Brinar 400 | 900 | 1200 | 12 | 25 (−20 °C) | 340–440 |
| Brinar 500 | 1350 | 1500 | 8 | 25 (−20 °C) | 480 |
| XAR 600 | 1700 | 2000 | 8 | 25 (−20 °C) | >550 |

$R_{p0.2}$—yield strength, $R_m$—ultimate tensile strength, $A_5$—percentage elongation after fracture for proportional specimens with the original gauge length $L_0$ equal to 5 times diameter, $KCV_{-40}$—Charpy V-notch toughness at −40 °C, *HBW*—Brinell hardness, NA—no available data.

**Table 2.** Chemical compositions and declared carbon equivalents of selected abrasive-wear resistant steels [1–5].

| Element | Hardox | | | | | Brinar | | XAR |
| | 400 | 450 | 500 | 600 | Extreme | 400 | 500 | 600 |
| | Selected Element [wt%] | | | | | | | |
|---|---|---|---|---|---|---|---|---|
| C | 0.32 | 0.26 | 0.30 | 0.45 | 0.47 | 0.18 | 0.28 | 0.40 |
| Mn | 1.60 | 1.60 | 1.60 | 1.40 | 1.40 | 2.00 | 1.50 | 1.50 |
| Si | 0.70 | 0.70 | 0.70 | 0.70 | 0.50 | 0.50 | 0.80 | 0.80 |
| P | 0.025 | 0.025 | 0.020 | 0.015 | 0.015 | 0.015 | 0.020 | 0.025 |
| S | 0.010 | 0.010 | 0.010 | 0.010 | 0.010 | 0.005 | 0.005 | 0.010 |
| Cr | 2.40 | 1.40 | 1.50 | 1.20 | 1.20 | 1.55 | 1.50 | 1.50 |
| Ni | 1.50 | 1.50 | 1.50 | 2.50 | 2.50 | NA | NA | 1.50 |
| Mo | 0.60 | 0.60 | 0.60 | 0.70 | 0.80 | 0.60 | 0.40 | 0.50 |
| B | 0.004 | 0.005 | 0.005 | 0.005 | 0.005 | 0.005 | NA | 0.005 |
| # [mm] | 8–20 | 10–19.9 | 4–13 | 6–35 | 8–19 | ≤80 | ≤60 | 15 |
| $CEV^T$ | 0.44 | 0.48 | 0.51 | 0.66 | 0.66 | NA | NA | 0.79 |
| $CET^T$ | 0.28 | 0.36 | 0.37 | 0.55 | 0.55 | NA | NA | 0.53 |

$CEV^T$—typical carbon equivalent according to International Institute of Welding, $CET^T$—typical carbon equivalent according to SS-EN 1011-2, #—sheet thickness for the given chemical properties, NA—no available data.

Development of the subject matter concerning welding of high-strength, abrasive-wear resistant martensitic steels seems to also be well-grounded in the context of numerous negative opinions of users of these steels, undertaking technological activities similar to these discussed here. These opinions predominantly result from big discrepancies between the data published in material data sheets and the results of our own examinations. The carbon equivalent value given by manufacturers is most often a typical value that is significantly different from that calculated on the grounds of real chemical composition of the given grade and sheet thickness. For this reason, activities undertaken by the users on the grounds of inaccurate data often do not provide positive results. This is why, in most cases, the use of the considered grades are given up in favor to their equivalents with much lower mechanical properties, but which are characterized by better carbon equivalents.

The performed examinations concerning the chemical and structural properties of low-alloyed abrasive-wear resistant steels [16,18–26] make it possible to formulate a general statement with regard to the good weldability of steels Hardox 400 and 450, as well as the satisfactory weldability of Hardox 500. This standpoint is also confirmed by the location of chemical compositions of these steels in the diagram *C-CEV* (Figure 1) close to the zones of low (I) or conditions-dependent (II) susceptibility to cracking. However, in the case of Hardox Extreme, this statement does not seem to be well-grounded (Table 3 and Figure 1) because of big discrepancies between the manufacturer's data (designation "P" in Figure 1) and own results (designation "O" in Figure 1). The discrepancy also concerns Hardox 600.

**Table 3.** Real chemical compositions and carbon equivalent values calculated on their grounds for sheet metal 8–15 mm thick of selected low-alloyed steels [18,24].

| Element | Hardox Steels | | | | | Brinar Steels | | XAR |
|---|---|---|---|---|---|---|---|---|
| | 400 | 450 | 500 | 600 | Extreme | 400 | 500 | 600 |
| | Selected Element [wt%] | | | | | | | |
| C | 0.17 | 0.17 | 0.28 | 0.45 | 0.48 | 0.17 | 0.28 | 0.37 |
| Mn | 1.00 | 1.00 | 0.69 | 0.51 | 0.52 | 1.14 | 0.95 | 0.85 |
| Si | 0.37 | 0.32 | 0.26 | 0.16 | 0.16 | 0.22 | 0.66 | 0.19 |
| P | 0.010 | 0.010 | 0.011 | 0.012 | 0.010 | 0.008 | 0.012 | 0.014 |
| S | 0.002 | 0.000 | 0.001 | 0.002 | 0.001 | 0.000 | 0.000 | 0.001 |
| Cr | 0.22 | 0.45 | 0.66 | 0.33 | 0.89 | 0.60 | 0.84 | 0.83 |
| Ni | 0.05 | 0.05 | 0.08 | 1.98 | 1.96 | 0.38 | 0.01 | 1.21 |
| Mo | 0.01 | 0.08 | 0.03 | 0.14 | 0.13 | 0.30 | 0.20 | 0.15 |
| V | 0.004 | 0.005 | 0.010 | 0.009 | 0.008 | 0.039 | 0.006 | 0.002 |
| Cu | 0.006 | 0.018 | 0.016 | 0.016 | 0.021 | 0.010 | 0.008 | 0.030 |
| Al | 0.035 | 0.032 | 0.050 | 0.031 | 0.034 | 0.073 | 0.039 | 0.097 |
| Ti | 0.020 | 0.016 | 0.005 | 0.006 | 0.006 | 0.009 | 0.012 | 0.003 |
| Nb | 0.010 | 0.000 | 0.000 | 0.005 | 0.001 | 0.043 | 0.023 | 0.009 |
| Co | 0.010 | 0.016 | 0.017 | 0.026 | 0.022 | 0.001 | 0.005 | 0.005 |
| B | 0.0016 | 0.0014 | 0.0016 | 0.0026 | 0.0025 | 0.0023 | 0.0008 | 0.0021 |
| # [mm] | 8 | 10 | 10 | 10 | 10 | 12 | 12 | 15 |
| *CEV* | 0.38 | 0.44 | 0.54 | 0.76 | 0.90 | 0.58 | 0.65 | 0.79 |
| *CET* | 0.28 | 0.30 | 0.39 | 0.58 | 0.64 | 0.36 | 0.43 | 0.54 |

*CEV* [%] = C + Mn/6 + (Cr + Mo + V)/5 + (Cu + Ni)/15; *CET* [%] = C + (Mn + Mo)/10 + (Cr + Cu)/20 + Ni/40, *CEV*—carbon equivalent according to International Institute of Welding, *CET*—carbon equivalent according to SS-EN 1011-2, #—sheet thickness for the given chemical properties.

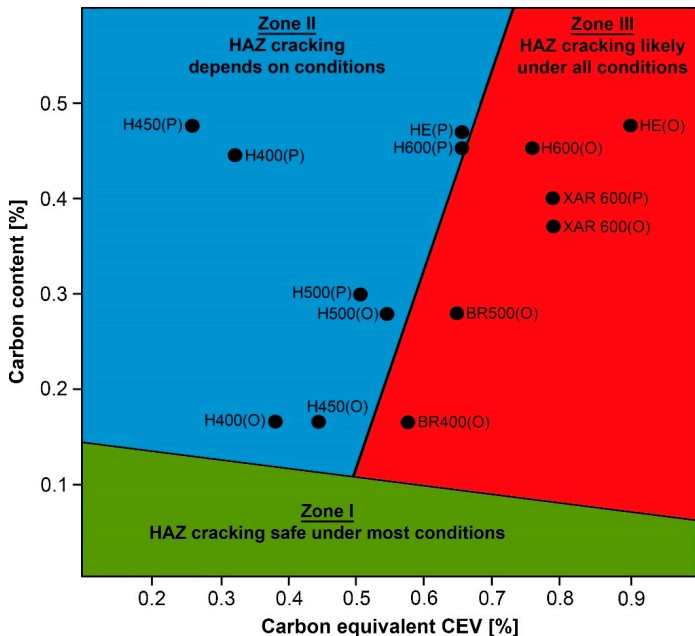

**Figure 1.** Susceptibility to cracking in function of carbon content and *CEV* of selected abrasive-wear resistant steels. H—Hardox, BR—Brinar, P—manufacturer's data, O—own results. Development based on data from Table 3 and [27].

The above-described discrepancies decidedly provide evidence against these steels with regard to their metallurgical weldability, shifting *CEV* values from the zone (II) to the zone (III) of high susceptibility to cracking in any welding conditions. According to users and manufacturers of abrasive-wear resistant steels, the most often indicated problems of weldability of Hardox Extreme (and also Hardox 600) are related to the susceptibility of the made welds to the brittle cracking (including also delayed cracking) and very wide zones with lower hardness in comparison to the base material. In this connection, the purpose of the presented research work is the identification of macro- and microscopic structures of Hardox Extreme welded joints in the as-welded condition (after welding) and determination of the area of structural changes within the entire welded joint, as well as provoking, through heat treatment, structural changes in order to eliminate or minimize the previously existing adverse structures. It is worth mentioninging that similar examinations of Hardox 600 were also carried out. However, with regard to the very comprehensive set of results, they will be elaborated on separately. It should be also mentioned that the author is currently conducting examinations of Hardox welded joints with regard to their abrasive wear in real conditions of soil abrasive mass.

## 2. Material and Methodology

Examinations were carried-out on Hardox Extreme steel sheets in an as-delivered condition, which was 1000 mm long and 10 mm thick. Welded joints were made by submerged arc welding *SAW* (121), while considering the welding materials dedicated for low-alloyed high-strength steels. Selected properties of the used welding materials are given in Table 4.

**Table 4.** Selected properties of welding materials used for Hardox Extreme joints [28].

| Weld Metal | C | Mn | Si | Cr | Ni | Mo | $R_{p0.2}$ | $R_m$ | $A_4$ | $KCV_{-40}$ |
| --- | --- | --- | --- | --- | --- | --- | --- | --- | --- | --- |
| | \multicolumn{6}{c}{Chemical Composition [%]} | | | | | | [MPa] | | [%] | [J/cm$^2$] |
| **OK Autrod 13.43 + OK Flux 10.62** | 0.11 | 1.50 | 0.25 | 0.60 | 2.20 | 0.50 | 700 | 800 | 21 | 94 |

$R_{p0.2}$—yield strength, $R_m$—ultimate tensile strength, $A_4$—percentage elongation after fracture for proportional specimens with the original gauge length $L_0$ equal to 4 times diameter, $KCV_{-40}$—Charpy V-notch toughness at −40 °C.

Joints were made using an automatic welding machine ESAB A2 Mini Trac with the power source ESAB LAE 800. The Hardox sheets were joined with a both-sides weld (Figure 2) using the following parameters guaranteeing correct penetration:

- weld type: BW (butt weld),
- welding position: PA (flat),
- electrode diameter: 3.0 mm,
- arc voltage (weld layer: 1, 2): 34/35 V,
- amperage (weld layer: 1, 2): 520/640 A,
- polarity: DC(+),
- welding rate (weld layer: 1, 2): 61/63 cm/min,
- electrode wire: OK Autrod 13.43 (S3Ni2.5CrMo acc. to EN ISO 26304),
- flux: OK Flux 10.62,
- preheating: no,
- interpass temperature: ≤100 °C,
- preparation of sheet edges (chamfering): no.

After welding, test specimens were cut-out in form of cuboids by means of a high-energy abrasive water stream (general specimen geometry) and through electroerosion (V-notch geometry). Next, a part of the specimens was subjected to heat treatment in laboratory conditions by being quenched in oil and tempering. It should be stressed that, with regard to the posed cognitive goals and available technical measures, the heat treatment operations were carried-out on whole specimens, i.e., for both heat-affected zone and base material. Before quenching, the specimens were additionally subjected to normalizing. All the thermal operations were carried-out in gas-tight chamber furnaces Czylok FCF 12SHM/R in a protective atmosphere of 99.95% argon. Quenching was carried-out in quenching oil Durixol W72 with a kinematic viscosity of 21 mm$^2$/s, heated-up to 50 ± 5 °C. Detailed characteristics of the specimens and heat treatment parameters are given in Table 5.

Chemical analyses were carried-out spectrally by means of a glow discharge spectrometer Leco GDS-500A, using the following parameters: $U$ = 1250 V, $I$ = 45 mA, 99.999% argon. The results were given as averages of at least five measurements.

Observations of macro- and micro-structures were performed using a multifunctional stereoscopic microscope Nikon AZ100 and a light microscope Nikon Eclipse MA200 coupled with a digital camera Nikon DS-Fi2. Images were recorded and analyzed using software NIS Elements.

Rockwell hardness (HRC/HRA) measurements were taken with a universal hardness tester Zwick/Roell ZHU 187.5 at 1500/600 kgf according to EN ISO 6508-1:2016-10. Measurements were taken on the specimens after microstructure examinations, within the base material (Hardox Extreme sheets) and in the zones subjected to structural analysis (lines A and B marked in Figure 2a).

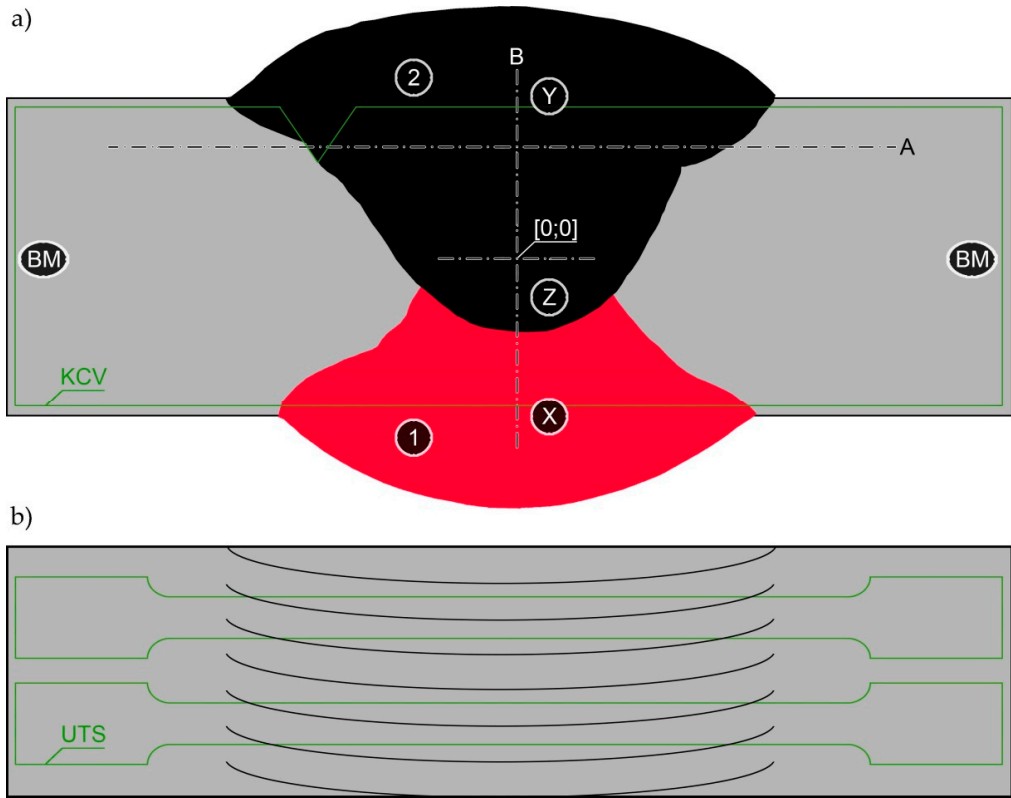

**Figure 2.** General layout of a Hardox Extreme welded joint: (**a**) cross-section view; (**b**) view from the face of the weld. 1,2—individual welds acc. to their execution order; A,B—lines of hardness measurements; BM—base material; X,Y,Z—places of chemical analyses; KCV—way of cutting out specimens for impact tests—dimensions of the specimen after V-notching: 8.5 × 10 × 55 mm; UTS—way of cutting out specimens for ultimate tensile strength—dimensions of the cuboidal specimens with preset gauge length of $L_0$ = 25 mm: 10 × 10 mm.

Mechanical tests were carried-out at ambient temperature according to EN ISO 6892-1:2016-09, using a testing machine Instron 5982, on cuboidal specimens with preset gauge length of $L_0 = 25$ mm (Figure 2b). The tensile tests were carried-out under controlled force to ensure a uniform strain rate for the specimens until their failure. Next, tensile strength ($R_m$) and reduction of area at failure ($Z$) were determined.

Impact tests of the welded joints were carried-out using a pendulum Charpy tester Zwick/Roell RPK300 with initial energy 300 J, according to EN ISO 148-1:2017-02. The V-notch specimens used in the tests were cut from the entire butt joints (according to EN ISO 9016:2011) and included fusion zones of the analyzed welded joints in the conditions directly after welding and after heat treatment operations (Figure 2a). The tests were carried-out at +20 °C and −40 °C. Observations of fracture surfaces were performed with a stereoscopic microscope and a scanning electron microscope JEOL JSM-6610A. The SEM observations were performed at an accelerating voltage of 20 kV, in material contrast to what occurs when using SE detectors.

## 3. Results

Basic parameters of heat treatment operations carried-out on the examined welded joints are given in Table 5 and Figures 3 and 4, together with results of mechanical and impact tests of the joints in the conditions directly after welding and after heat treatment operations.

The performed heat treatment operations were aimed at obtaining a microstructure and mechanical properties in the entire welded joints that are similar to those of the base material. As such, the welded joints were volumetrically quenched in oil bath and tempered (stress relieved). The austenitizing temperature before quenching was established on the grounds of real chemical compositions of both the base material (Table 3) and weld metal, while considering complete cross-section of the welded joint (Table 6). Establishing the tempering temperature at 100 °C resulted from the fact that exposure of Hardox Extreme to temperatures over 125 °C causes decomposition of martensitic structure and significant decrease of hardness. This results also in lower indices from static tensile test and can lead to lower abrasive-wear resistance.

**Table 5.** Heat treatment parameters and selected mechanical properties of Hardox Extreme welded joint:

| Specimen | Heat Treatment Parameters | $R_m$ [MPa] | | $Z$ [%] | | $KCV_{+20}$ | | $KCV_{-40}$ [J/cm$^2$] | |
|---|---|---|---|---|---|---|---|---|---|
| UTS-1 | | 1329 | | 26.0 | | | | | |
| UTS-2 | | 1263 | 1278 | 16.3 | 17.4 | | | — | |
| UTS-3 | | 1242 | | 10.1 | | | | | |
| KCV-1 | | | | | | 18.7 | | | |
| KCV-2 | No treatment | | | | | 16.4 | 17 | | |
| KCV-3 | | | — | | | 16.6 | | | |
| KCV-4 | | | | | | | | 14.3 | |
| KCV-5 | | | | | | — | | 12.7 | 18 |
| KCV-6 | | | | | | | | 26.1 | |
| UTS-25 | Normalization: 800 °C/1h/Air + Quenching: 850 °C/20′/Oil + Tempering: 100 °C/20h/Air | 1831 | | 30.6 | | | | | |
| UTS-26 | | 1823 | 1831 | 31.6 | 29.1 | | | — | |
| UTS-28 | | 1839 | | 25.2 | | | | | |
| KCV-7 | | | | | | 27.1 | | | |
| KCV-8 | | | | | | 27.9 | 27 | | |
| KCV-9 | | | — | | | 25.9 | | — | |
| KCV-10 | | | | | | | | 21.6 | |
| KCV-11 | | | | | | — | | 17.1 | 19 |
| KCV-12 | | | | | | | | 19.0 | |

$R_m$—ultimate tensile strength, $Z$—percentage reduction of area, $KCV_{+20}$—charpy V-notch toughness at room temperature, $KCV_{-40}$—charpy V-notch toughness at −40 °C.

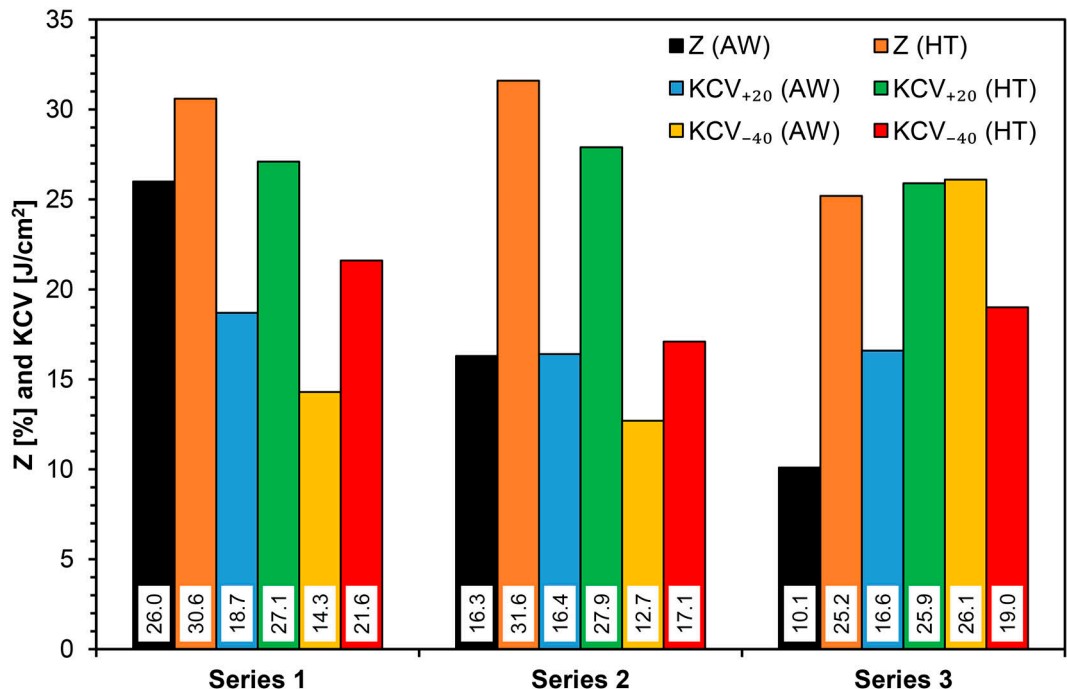

**Figure 3.** Percentage reduction of area *Z* and toughness *KCV* of tested samples based on data from Table 5. AW—condition after welding, HT—condition after heat treatment.

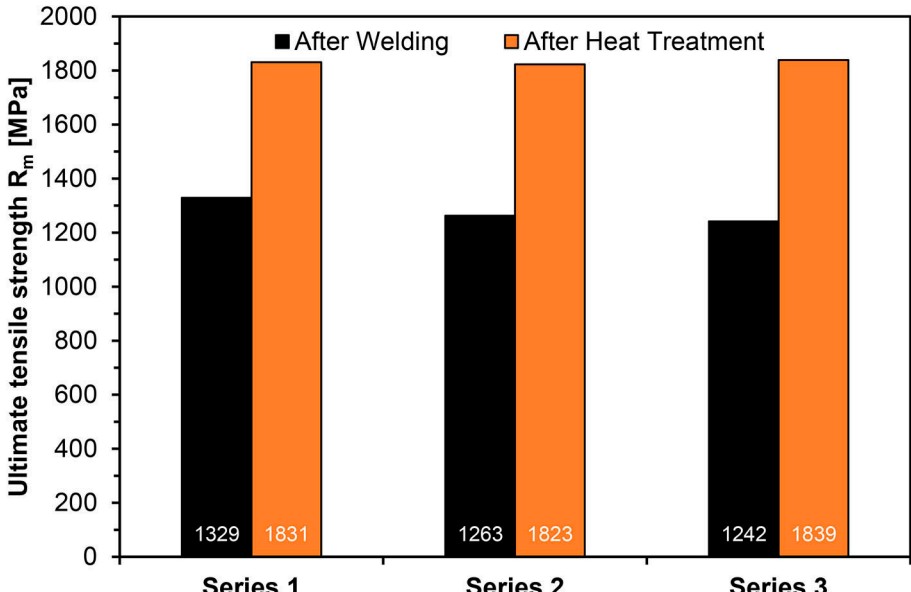

**Figure 4.** Ultimate tensile strength $R_m$ of tested samples based on data from Table 5.

**Table 6.** Chemical composition in cross-section of the welded joint.

| Element | X | Y | Z | OK 13.43 + OK Flux 10.62 | Hardox Extreme |
|---|---|---|---|---|---|
| | **Chemical Composition [wt%]** | | | | |
| C | 0.31 | 0.32 | 0.31 | 0.10 | 0.48 |
| Mn | 0.81 | 0.82 | 0.81 | 1.22 | 0.52 |
| Si | 0.23 | 0.23 | 0.23 | 0.30 | 0.16 |
| P | 0.013 | 0.013 | 0.014 | 0.020 | 0.010 |
| S | 0.002 | 0.002 | 0.002 | 0.002 | 0.001 |

**Table 6.** *Cont.*

| Element | X | Y | Z | OK 13.43 + OK Flux 10.62 | Hardox Extreme |
|---|---|---|---|---|---|
| | Chemical Composition [wt%] | | | | |
| Cr | 0.81 | 0.82 | 0.81 | 0.50 | 0.89 |
| Ni | 2.06 | 2.08 | 2.07 | 1.65 | 1.96 |
| Mo | 0.23 | 0.23 | 0.22 | 0.31 | 0.13 |
| V | 0.009 | 0.009 | 0.010 | 0.009 | 0.008 |
| Cu | 0.040 | 0.038 | 0.041 | 0.105 | 0.021 |
| Al | 0.020 | 0.021 | 0.020 | 0.013 | 0.034 |
| Ti | 0.004 | 0.004 | 0.005 | 0.003 | 0.006 |
| Nb | 0.000 | 0.000 | 0.000 | 0.000 | 0.001 |
| Co | 0.017 | 0.016 | 0.018 | 0.006 | 0.022 |
| B | 0.0021 | 0.0021 | 0.0021 | 0.0014 | 0.0025 |
| *CEV* | 0.79 | 0.80 | 0.80 | 0.59 | 0.90 |
| *CET* | 0.51 | 0.51 | 0.51 | 0.33 | 0.64 |

*CEV* [%] = C + Mn/6 + (Cr + Mo + V)/5 + (Cu + Ni)/15; *CET* [%] = C + (Mn + Mo)/10 + (Cr + Cu)/20 + Ni/40.3.1. X,Y,Z—places of chemical analyses, marked in Figure 2, *CEV*—carbon equivalent according to International. Institute of Welding, *CET*—carbon equivalent according to SS-EN 1011-2.

### 3.1. Mechanical Properties

Average tensile strength of Hardox Extreme welded joints in the condition after welding reached $R_\mathrm{m}$ = 1278 MPa (Table 5 and Figure 4), with maintained moderate plastic properties defined by relative area reduction $Z$ = 17.4% (Table 5 and Figure 3). Even if the obtained strength is very high in comparison to that of constructional steels, it only provides ca. 64% of the value for the base material (assuming minimum tensile strength of Hardox Extreme equal to that of Hardox 600, i.e., 2000 MPa). Here, a significant scatter of the obtained values should be indicated. Even if the test joints were made on an automated welding station with significant lengths of run-off welds of over 150 mm, the results were characterized by rather low repeatability. In the author's opinion, such a behavior of these materials after welding is caused by uncontrolled structural changes occurring in the heat-affected zone, which is the probable cause of the unpredictable behavior of these steels in industrial conditions. This observation can be related mostly to plastic properties of the weld, which, to some extent, is also confirmed by relatively low, widely scattered impact strength values (especially at negative testing temperatures). Average impact strength at both testing temperatures of 17–18 J/cm$^2$ clearly shows the necessity to classify the obtained welded joints as being susceptible to brittle cracking.

As a result of the applied thermal treatments, all the considered mechanical properties of Hardox Extreme welded joints increased significantly (Table 5). Average tensile strength amounted to 1831 MPa, which makes nearly 90% of the assumed strength of the base material. In addition to very high strength indices, the plastic properties increased. The average value of relative area reduction at failure was $Z$ = 29.1%, and impact energy occurred at an ambient temperature amounting to 27 J/cm$^2$, allowing the supposition that ductile fracture provides a significant part of the entire fracture area.

### 3.2. Results of Microscopic Examinations and Hardness Measurements

Figures 5–23 show macroscopic images, hardness measurements and an overview of characteristic microstructures in the entire area of the welded joint. It should be stated on the grounds of the performed examinations that the welding process provoked in the considered steel diverse structural changes that resulted in wide zones with decreased hardness, generally designated as heat-affected zones (Figures 5 and 7). Width of this zone was mainly decided by the used technology, parameters and conditions of welding, the value of delivered heat input, chemical composition of steel, and also by the structure of the base material before welding (Figure 9). High influence of the latter parameter on properties of the welded joint can be explained by tempering the processes of post-martensitic structures occurring mainly in the heat-affected zone. In the case of Hardox Extreme steel, the entire

zone affected by the temperature exceeding 125 °C is characterized by wide areas of reduced hardness (Figure 7). Experiences of the author indicate that, in many cases, the width of this zone exceeds 60–100 mm. In turn, application of high-energy welding methods to a steel with chemical composition close to that of Hardox Extreme very often results in cold cracking and reduced brittleness threshold determined in impact tests.

Conclusions similar to these above cannot be formulated in relation to the specimens after heat treatment. In the macroscopic image of the joint in this condition (Figure 6) no heat-affected zone is observed, which is additionally confirmed by hardness measurements (Figures 7 and 8). It should be also stressed that, from the viewpoint of real chemical composition of Hardox Extreme (*CEV* = 0.90; Table 3 and Figure 1), application of generally accepted weldability criteria to low-alloyed steels can give rise to serious doubts. In this connection, the author believes that, with regard to very high mechanical properties of the base material, operations of post-weld heat treatment should be obligatorily considered at welding processes of the steel Hardox Extreme. Such an approach makes it possible to restore microstructure and mechanical properties of the entire area of a welded joint.

Analysis of structural changes of the examined welded joints showed, in the zone of base material after welding (BM in Figure 5), microstructure characteristic for toughening processes, i.e., tempered sorbite (Figure 9b). This structure shows the tempering processes occurring in the welded material, resulting in significant loss of hardness in comparison to the condition before welding (Figure 7). The carried-out thermal operations made it possible to obtain, in the analogous zone, structures very well corresponding with the Hardox base material in an as-delivered condition. After heat treatment, the base material zone (BM in Figure 6) showed microstructure of fine-lath hardening martensite, almost identical to that of Hardox Extreme in as-delivered condition (Figure 9a,c). The above finding is confirmed by hardness measurements taken in the considered zones. After welding, level of hardness in the BM zone reached 320–350 HV (Figure 7). Heat treatment resulted in an increase of hardness to over 650 HV (Figure 7), which only slightly declines from the value declared by the manufacturer of Hardox Extreme in an as-delivered condition, i.e., ca. 700 HV (min. 650 HBW, see Table 1).

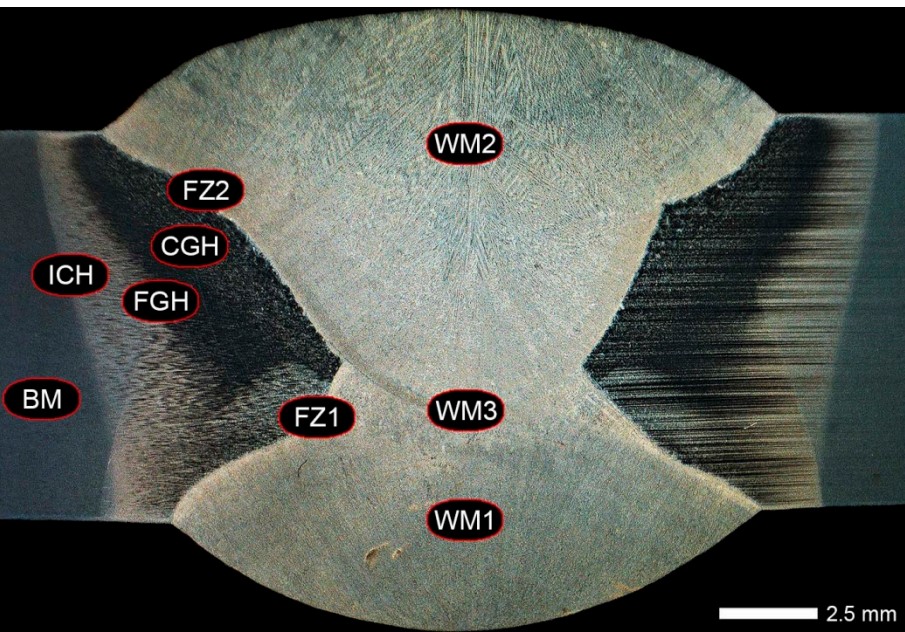

**Figure 5.** Macroscopic image of cross-section of a Hardox Extreme welded joint after welding. WM—weld metal zone, FZ—fusion zone, BM—base material zone, CGH(AZ)—coarse-grained heat-affected zone (overheating), FGH(AZ)—fine-grained heat-affected zone (normalization and recrystallisation), ICH(AZ)—intercritical heat-affected zone (incomplete normalization). Stereoscopic microscopy, etched with 3% $HNO_3$ and Adler's etchant.

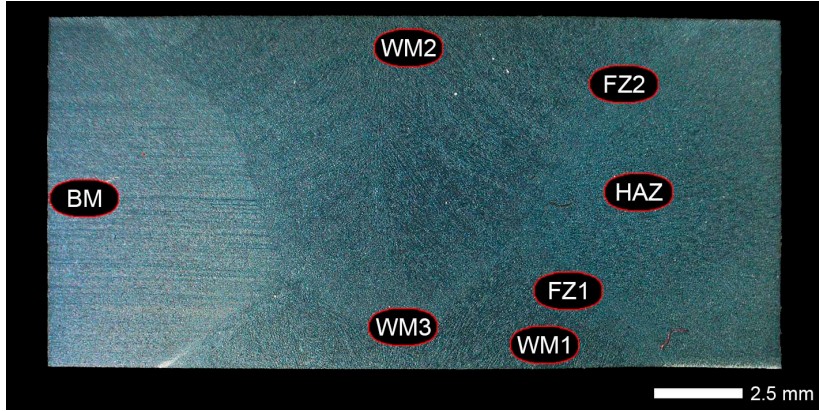

**Figure 6.** Macroscopic image of cross-section of a Hardox Extreme welded joint after heat treatment. WM—weld metal zone, FZ—fusion zone, BM—base material zone, HAZ—heat-affected zone. Stereoscopic microscopy, etched with 3% $HNO_3$.

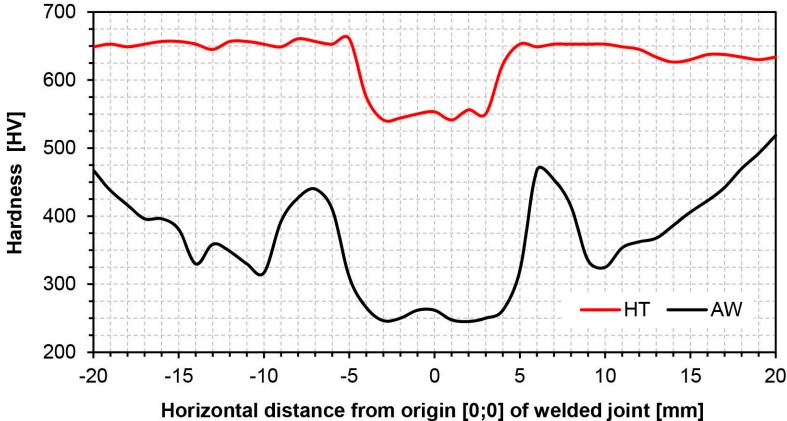

**Figure 7.** Hardness distribution of a Hardox Extreme welded joint along the line A shown in Figure 2. AW—condition after welding, HT—condition after heat treatment.

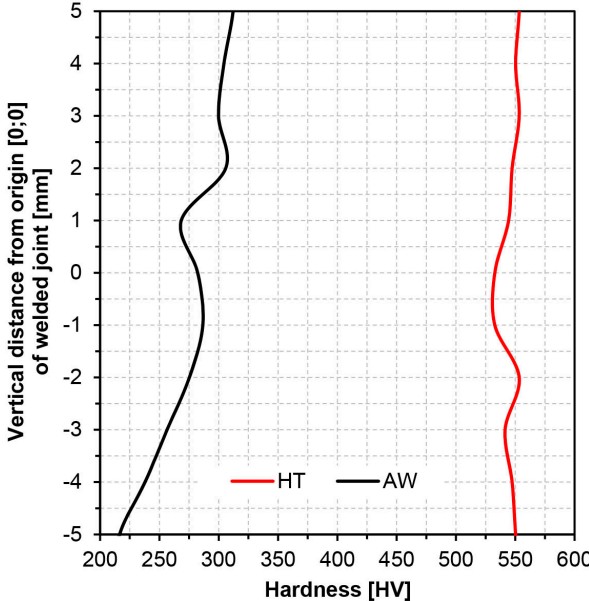

**Figure 8.** Hardness distribution of a Hardox Extreme welded joint along the line B shown in Figure 2. AW—condition after welding, HT—condition after heat treatment.

Hardness level in the weld metal zone (WM1, WM2 and WM3 in Figure 6) after heat treatment was ca. 550 HV, slightly lower than that of the base material (Figure 7). A drop of hardness by ca. 100 HV can be explained by slightly lower percentage of carbon in the weld axis, which decides saturation degree of ferrite and thus hardening capacity of the material. However, it is worth mentioninging that the obtained hardness on the entire thickness of the heat-treated welded joint was nearly linear at the level of 550 HV (Figure 8). This feature clearly distinguishes this material condition from that after welding, where hardness along the weld thickness changes within 220 to 310 HV, which also determines individual usable properties of the joint depending on sequence of applying the weld layers. Structural examinations indicate that the above conclusions concerning hardness should be related to microstructure changes in representative areas of the joint. In the zones designated WM1, WM2 and WM3 in Figure 5, microstructure after welding was typical for diverse temperatures and cooling rates. Generally, the microstructure of the weld metal is of a dendritic nature and composes of bands of martensite and small quantity of bainite on the background of non-equilibrium ferrite grains with features of a Widmanstätten structure in the WM1 zone (Figure 10).

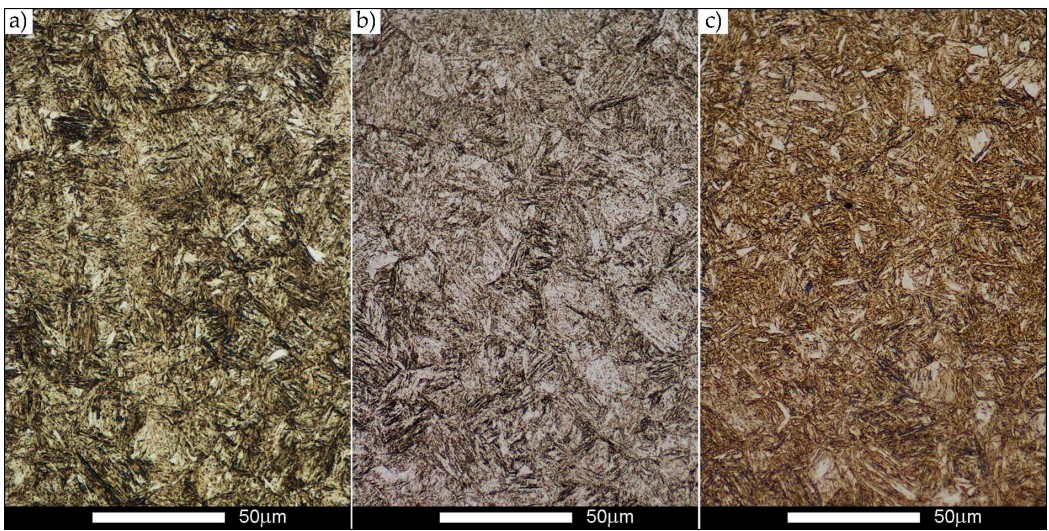

**Figure 9.** Microstructure of base material Hardox Extreme: (**a**) condition before welding—structure of tempered martensite; (**b**) condition after welding—BM zone in Figure 5, characteristic structure of tempered sorbite; (**c**) condition after heat treatment—BM zone in Figure 6, structure of hardening martensite. Light microscopy, etched with 2% $HNO_3$.

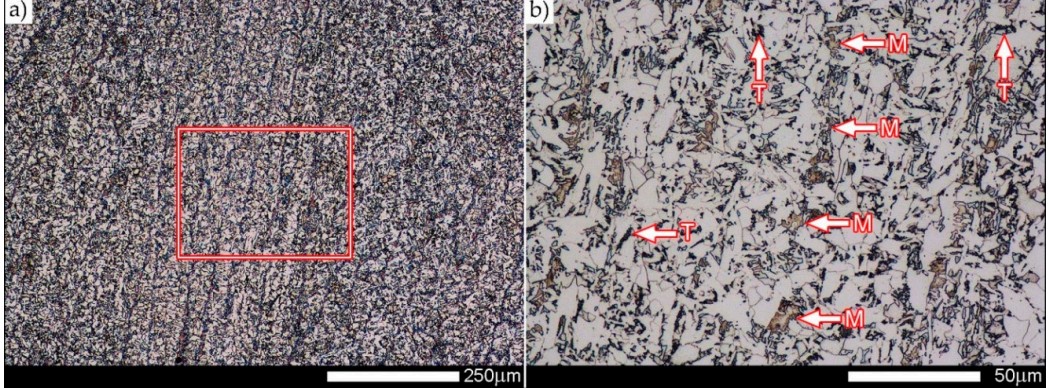

**Figure 10.** Microstructure of Hardox Extreme welded joint after welding: (**a**) in the area marked WM1 in Figure 5; (**b**) an enlarged image marked with frame in Figure 10a. Morphologically diverse structure composed of bands of martensite (M) with colonies of hardening troostite (T) on the background of non-equilibrium grains of ferrite. Light microscopy, etched with 2% $HNO_3$.

Microstructure in the zone WM2 is composed of lath hardening martensite on the background of tempered sorbite, small quantity of bainite and colonies of troostite (Figure 11). Microstructure in the transition zone between the weld layers 1 and 2 (WM3 in Figure 5) is composed mostly of hardening sorbite with band-like martensite, hardening bainite and colonies of troostite (Figure 12). It is also worth to indicate the significant change of structure in the zone WM3, which can determine general impact strength of the entire welded joint. Thus, from the viewpoint of fracture mechanics, the microstructure of the transition zone between the weld layers 1 and 2 (Figure 2) should be taken into account during the selection of technology and welding parameters for this type of metallic materials.

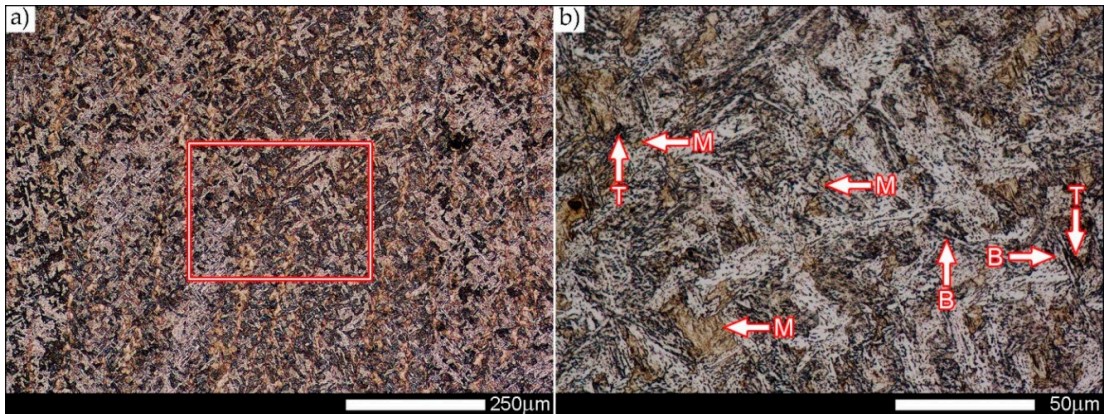

**Figure 11.** Microstructure of Hardox Extreme welded joint after welding: (**a**) in the area marked WM2 in Figure 5; (**b**) an enlarged image marked with frame in Figure 11a. Strongly diversified structure with dendritic nature, composed of bands of martensite (M) and bainite (B) and few colonies of troostite (T) on the grounds of hardening sorbite. Light microscopy, etched with 2% $HNO_3$.

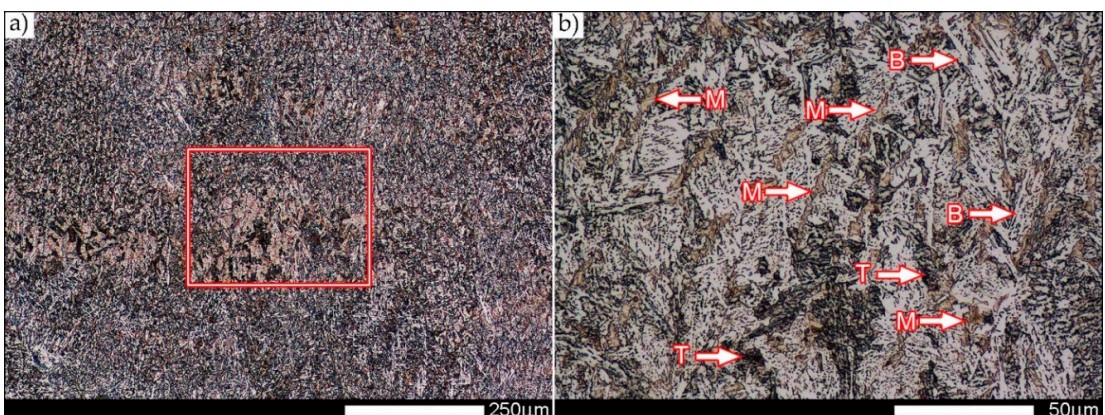

**Figure 12.** Microstructure of Hardox Extreme welded joint after welding: (**a**) in the area marked WM3 in Figure 5; (**b**) an enlarged image marked with frame in Figure 12a. Clearly visible, morphologically diversified fusion lines 1 and 2 composed of bands of martensite (M) and bainite (B) and colonies of troostite (T) on the background of hardening sorbite. Light microscopy, etched with 2% $HNO_3$.

After heat treatment, in all zones of the additional material (WM1, WM2 and WM3 in Figure 6), very similar microstructures (corresponding with that of as-delivered base material) were obtained, composed of fine-lath hardening martensite (Figures 13–15). With regard to the applied technological operations and their parameters, a remainder of band-like structure after welding can be observed in the recorded microstructures. Nevertheless, in the context of industrial applications, its complete removal would not be technologically and economically justified. However, it is worth mentioning that the applied heat treatment practically eliminated the observed structure changes in the zone marked WM3 in Figure 15.

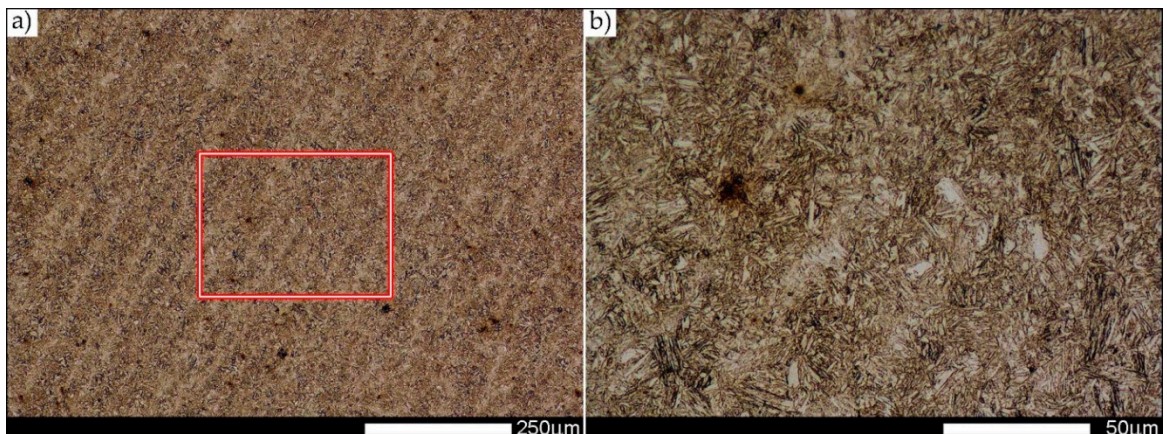

**Figure 13.** Microstructure of Hardox Extreme welded joint after heat treatment: (**a**) in the area marked WM1 in Figure 6; (**b**) an enlarged image marked with frame in Figure 13a. Structure of fine-lath hardening martensite with clear banding features resulting from former dendritic structure. Light microscopy, etched with 2% HNO$_3$.

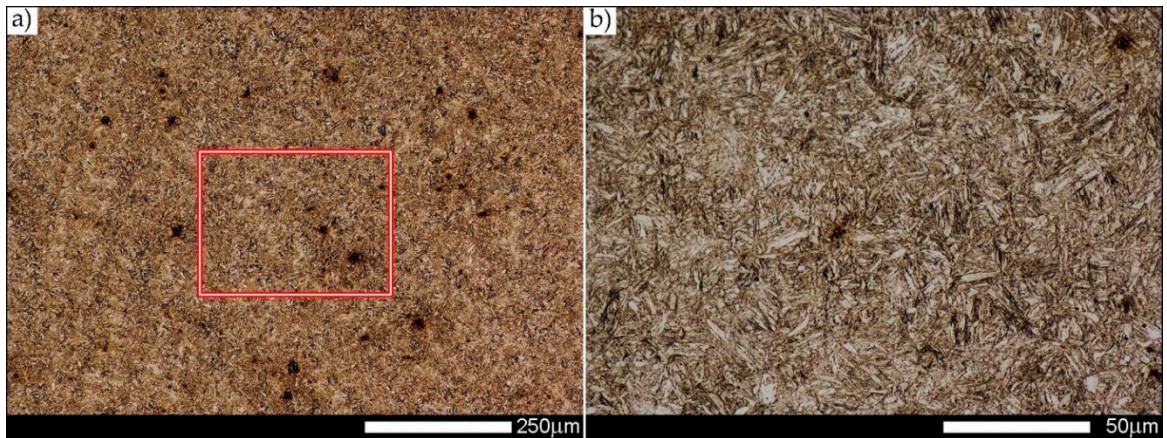

**Figure 14.** Microstructure of Hardox Extreme welded joint after heat treatment: (**a**) in the area marked WM2 in Figure 6; (**b**) an enlarged image marked with frame in Figure 14a. Structure of fine-lath hardening martensite. Light microscopy, etched with 2% HNO$_3$.

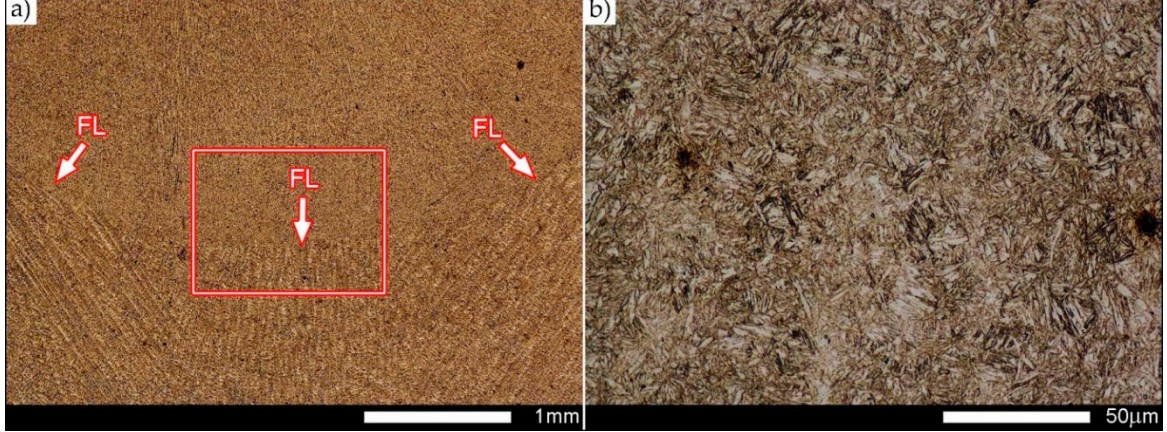

**Figure 15.** Microstructure of Hardox Extreme welded joint after heat treatment: (**a**) in the area marked WM3 in Figure 6; (**b**) an enlarged image marked with frame in Figure 15a. Structure of fine-lath hardening martensite with weak banding features resulting from former dendritic structure. Faintly outlined fusion line between weld layers 1 and 2 (FL) is marked with arrows. Light microscopy, etched with 2% HNO$_3$.

Observations of microstructures in the fusion zones indicate a very wide range of their kinds and morphologies. It can be generally found that the zones marked FZ1 and FZ2 in Figure 5 are characterized by a very clearly outlined fusion line (FL in Figures 16 and 17) composed of coarse-lath hardening martensite, areas of upper bainite and non-equilibrium grains of ferrite, and also hardening sorbite and troostite (Figures 16 and 17). In addition, the recorded microstructures are characterized by strongly heterogeneous morphology, even within the same type of structure. It should be stressed that creation of a complete characteristic of structures in the considered Hardox Extreme welded joint requires examinations with the use of transmission electron microscopy (TEM) that is currently being carried-out.

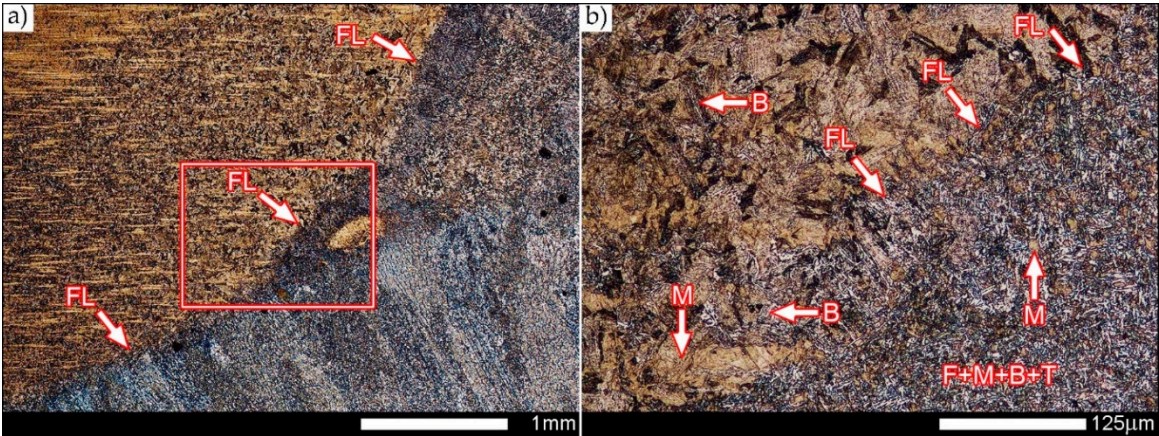

**Figure 16.** Microstructure of Hardox Extreme welded joint after welding: (**a**) in the area marked FZ1 in Figure 5; (**b**) an enlarged image marked with frame in Figure 16a. Very clearly outlined fusion line (FL) with strongly diversified microstructure. Visible islands of hardening martensite (M) and bainite (B), troostite areas (T) and acicular ferrite (F). Light microscopy, etched with 2% HNO$_3$.

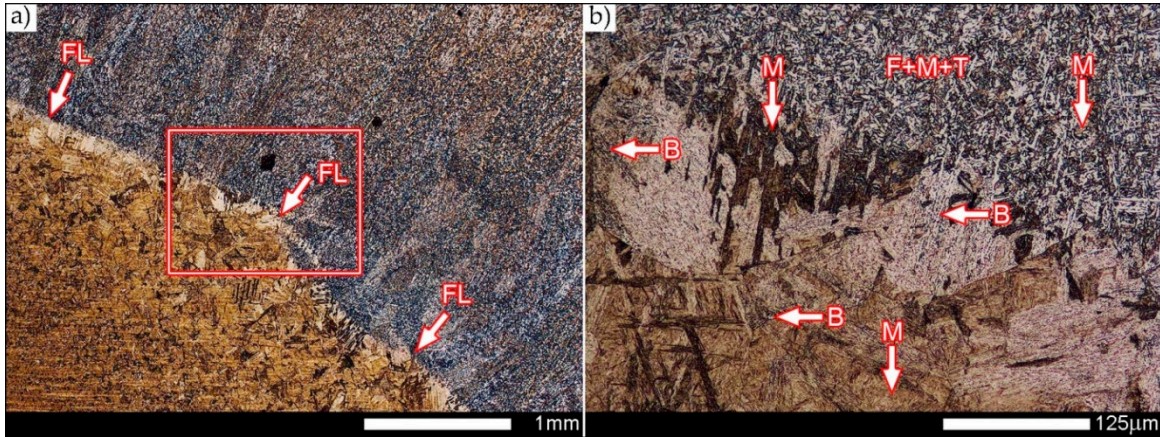

**Figure 17.** Microstructure of Hardox Extreme welded joint after welding: (**a**) in the area marked FZ2 in Figure 5; (**b**) an enlarged image marked with frame in Figure 17a. Very clearly outlined fusion line (FL) with strongly diversified microstructure. Visible islands of hardening martensite (M) and bainite (B), small number of troostite colonies (T) and acicular ferrite (F). Light microscopy, etched with 2% HNO$_3$.

The performed heat treatment operations led to homogeneous structure in the fusion zones (FZ1 and FZ2 in Figure 6), with respect to both type and fineness. In both considered fusion zones, structures of fine-lath hardening martensite were observed on the very weakly outlined fusion line (Figures 18 and 19). The above findings are also confirmed by hardness measurements (Figure 7) that do not show clear differences in comparison to the joint after welding.

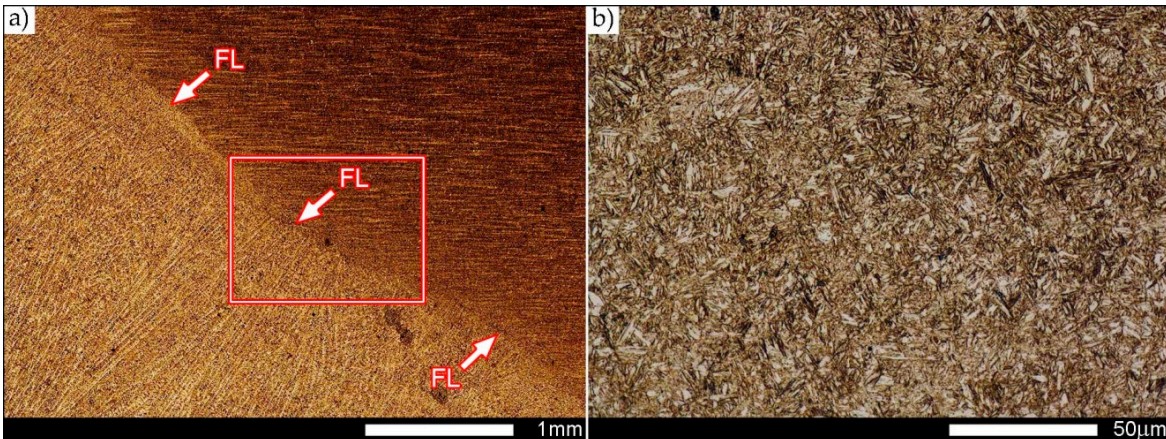

**Figure 18.** Microstructure of Hardox Extreme welded joint after heat treatment: (**a**) in the area marked FZ1 in Figure 6; (**b**) an enlarged image marked with frame in Figure 18a. Structure of fine-lath hardening martensite. The arrows (FL) indicate the very weakly outlined fusion line. Light microscopy, etched with 2% HNO$_3$.

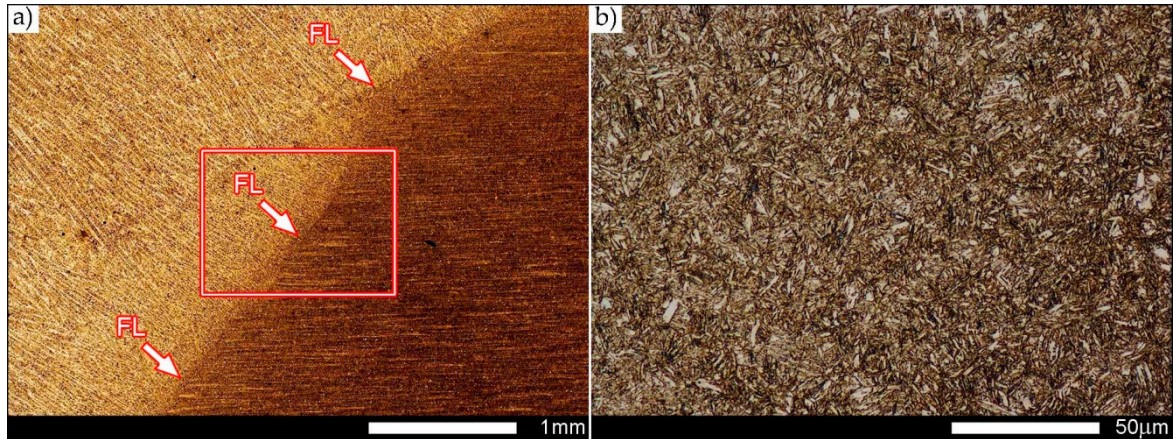

**Figure 19.** Microstructure of Hardox Extreme welded joint after heat treatment: (**a**) in the area marked FZ2 in Figure 6; (**b**) an enlarged image marked with frame in Figure 19a. Structure of fine-lath hardening martensite. The arrows (FL) indicate the very weakly outlined fusion line. Light microscopy, etched with 2% HNO$_3$.

Analysis of the other characteristic areas of heat-affected zone in the condition after welding, i.e., CGHAZ, FGHAZ and ICHAZ (marked in Figure 5), indicates much diversified structures in each case, resulting in significantly varying hardness levels. In the coarse-grained heat-affected zone (CGHAZ in Figure 20), fine- and coarse-lath hardening martensite is observed, with bainite separated on grain boundaries of former austenite, and also small quantity of hardening sorbite. In the fine-grained heat-affected zone (FGHAZ in Figure 21), microstructure includes mostly fine-lath hardening martensite with colonies of troostite and small number of bainitic areas. The intercritical heat-affected zone (ICHAZ in Figure 22) is composed of very fine, band-like arranged martensite with tempered sorbite. The morphology of this zone is clearly different than that of the base material, composed of fine-lath martensite (left side in Figure 22a,b).

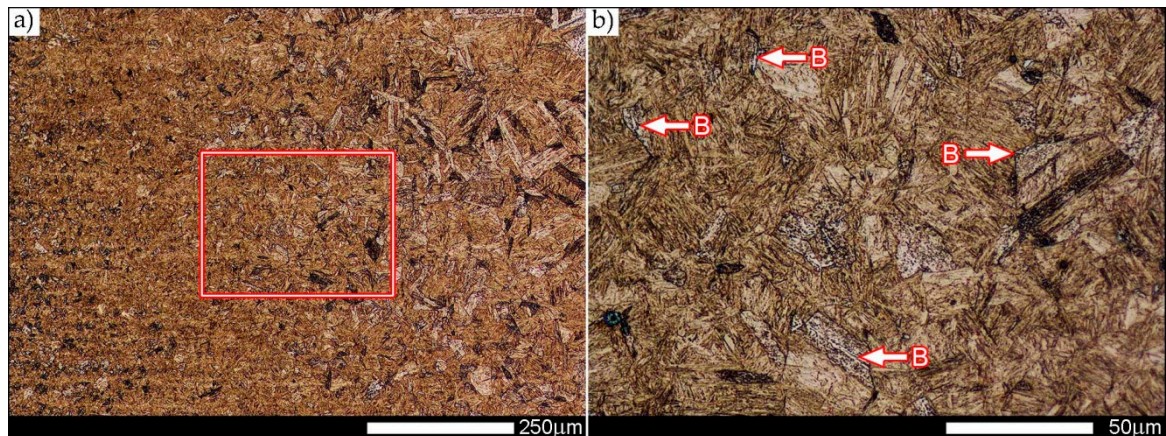

**Figure 20.** Microstructure of Hardox Extreme welded joint after welding: (**a**) in the area marked CGHAZ in Figure 5; (**b**) an enlarged image marked with frame in Figure 20a. Lath-like and locally acicular martensitic structure with bainite (B) on grain boundaries of former austenite. Light microscopy, etched with 2% $HNO_3$.

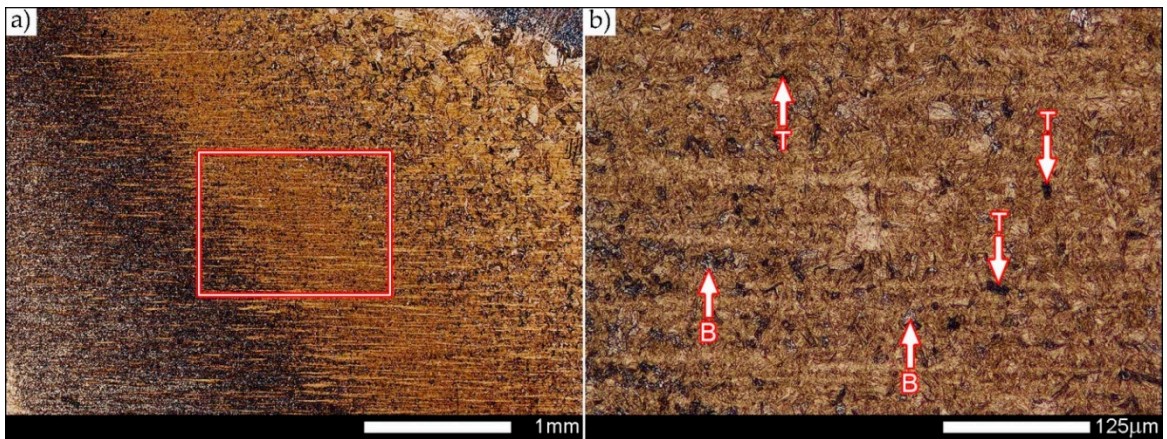

**Figure 21.** Microstructure of Hardox Extreme welded joint after welding: (**a**) in the area marked FGHAZ in Figure 5; (**b**) an enlarged image marked with frame in Figure 21a. Structure of fine-lath hardening martensite with inclusions of bainite (B) and troostite (T). Light microscopy, etched with 2% $HNO_3$.

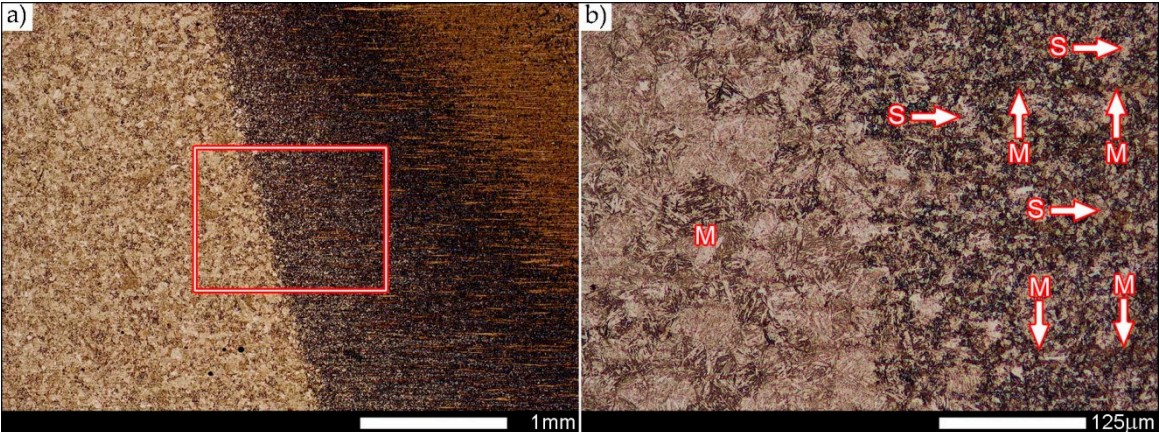

**Figure 22.** Microstructure of Hardox Extreme welded joint after welding: (**a**) in the area marked ICHAZ in Figure 5; (**b**) an enlarged image marked with frame in Figure 22a. Structure of fine-lath band-like arranged hardening martensite (M) with hardening sorbite (S). Light microscopy, etched with 2% $HNO_3$.

The performed heat treatment operations of the joint resulted in uniform microstructure in the entire heat-affected zone. The microstructure of this zone (HAZ in Figure 6) is composed of fine-lath hardening martensite with maintained banding features resulting from the process of thermo-mechanical rolling of Hardox sheet metal (Figure 23). As such, the hardness course in this zone is nearly linear at the average level of 650 HV (Figure 7). It can be stated on this ground that heat treatment of the joint brought the structure of heat-affected zone to the structure of the base material in the as-delivered condition.

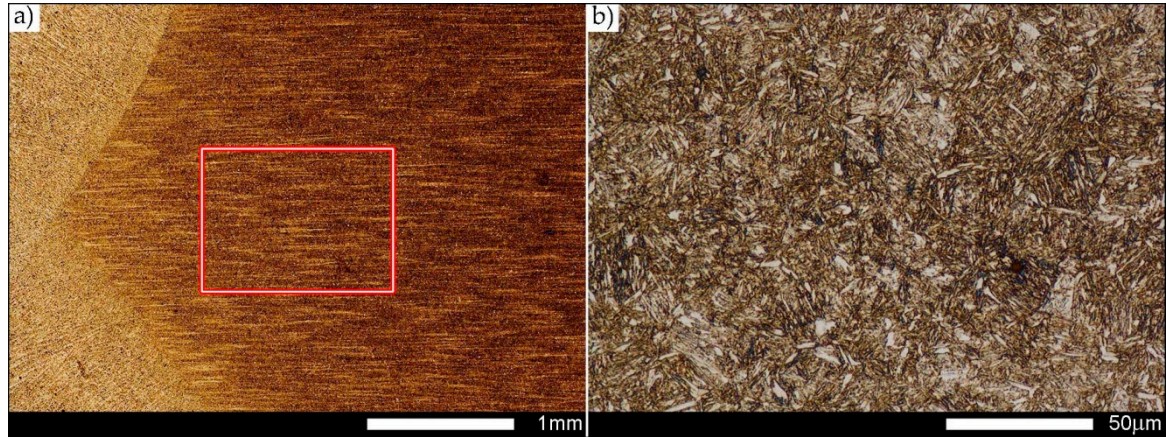

**Figure 23.** Microstructure of Hardox Extreme welded joint after heat treatment: (**a**) in the area marked HAZ in Figure 6; (**b**) an enlarged image marked with frame in Figure 23a. Structure of fine-lath hardening martensite. Light microscopy, etched with 2% HNO$_3$.

### 3.3. Results of Fractographic Analysis

Figures 24–28 show representative images of fracture surfaces of Hardox Extreme welded joints in conditions both after welding and after heat treatment operations. Fractographic analyses were made at the temperatures of impact testing, i.e., +20 °C and −40 °C. In each of the analyzed cases, fractures after impact tests do not show a significant share of plastic side zones (Figure 24), which proves relatively small energy expenditure during their creation.

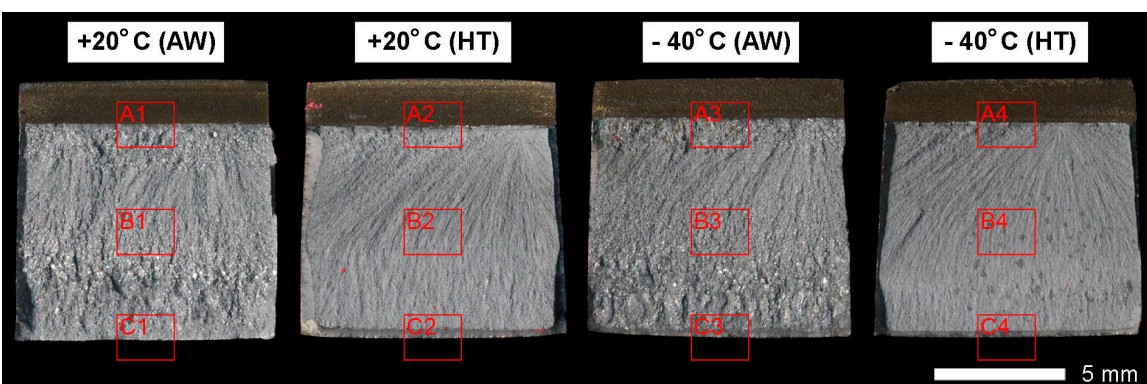

**Figure 24.** Macroscopic images of representative specimens of Hardox Extreme welded joints after impact testing. AW—condition after welding, HT—condition after heat treatment. Marked with frames: A—under-notch zone, B—central zone, C—final fracture zone. Stereoscopic microscopy, unetched.

This statement especially refers to fractures of the joint after welding, which were subjected to impact testing at the negative temperature. In addition, macroscopic analysis showed that fractures of the specimens after welding are characterized by highly diversified surface topography, resulting from the presence of coarse-grained structure on the fusion line, partially under the notch (frames A1 and A3 in Figure 24), in the central zone (frames B1 and B3 in Figure 24) and in major part of the

final fracture zone (frames C1 and C2 in Figure 24). In turn, fracture surfaces of the heat treated joint can be found to be uniform and rough on their entire area, showing a characteristic run according to the crystallization direction after welding (Figure 24). In order to reveal a detailed structure of individual zones, all fracture surfaces were subjected to further examinations by using scanning electron microscopy.

Transcrystalline fractures of Hardox Extreme welded joints in the condition after welding, subjected to examination at both ambient and reduced temperatures, are fractures of mixed nature, with irregularities on the separation surface (steps) and with a clearly visible "river" system (Figures 25 and 26).

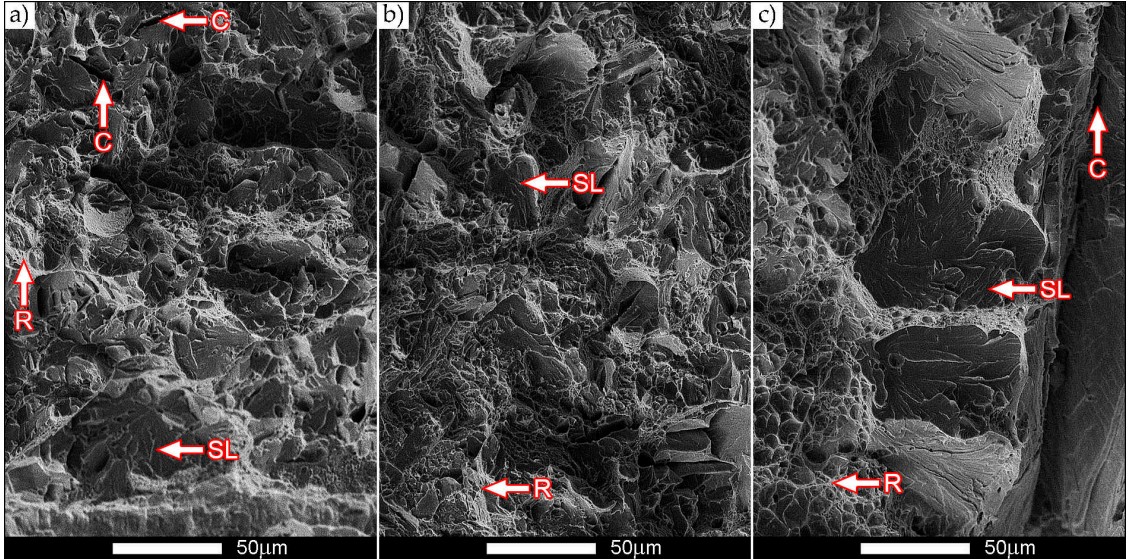

**Figure 25.** SEM images of fracture surfaces of Hardox Extreme welded joints after welding, shown in Figure 24, after impact testing at +20 °C. (**a**) Area marked with the frame A1; (**b**) area marked with the frame B1; (**c**) area marked with the frame C1. R—"river" system; SL—slides; C—microcracks with fine steps. Scanning microscopy, unetched.

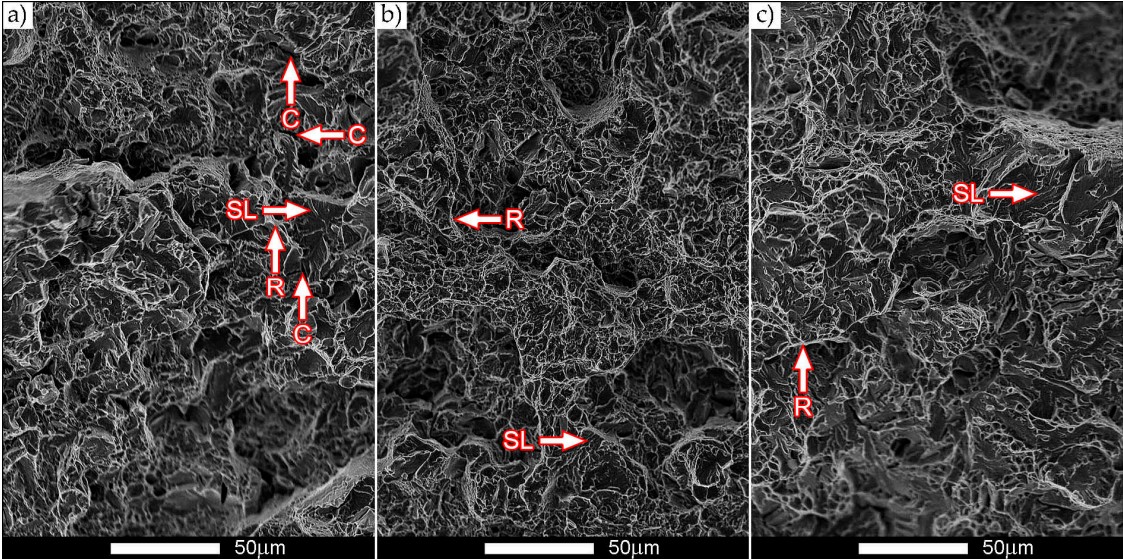

**Figure 26.** SEM images of fracture surfaces of Hardox Extreme welded joints after welding, shown in Figure 24, after impact testing at −40 °C. (**a**) Area marked with the frame A3; (**b**) area marked with the frame B3; (**c**) area marked with the frame C3. R—"river" system; SL—slides; C—microcracks with fine steps. Scanning microscopy, unetched.

In both cases, structures with micro-voids of different sizes can be also distinguished on fracture surfaces, where no inclusions of phases derived from alloying microadditives are observed. A characteristic feature of fracture surfaces of the specimens after welding is presence of numerous transverse microcracks with agglomerations of fine steps. Such a state is observed mostly in the zone under the notch (Figures 25a and 26a) and locally in the final fracture zone (Figure 25c). Moreover, almost in all areas of the fractures not subjected to heat treatment, slides are visible, which is characteristic for cleavage fractures.

Qualitative differences in fracture structures are clearly demonstrated on the specimens subjected to heat treatment after welding. On the surfaces, areas of micro-voids are observed, separated by plastic areas with band-like arrangement of "scaly" steps (Figures 27b,c and 28c).

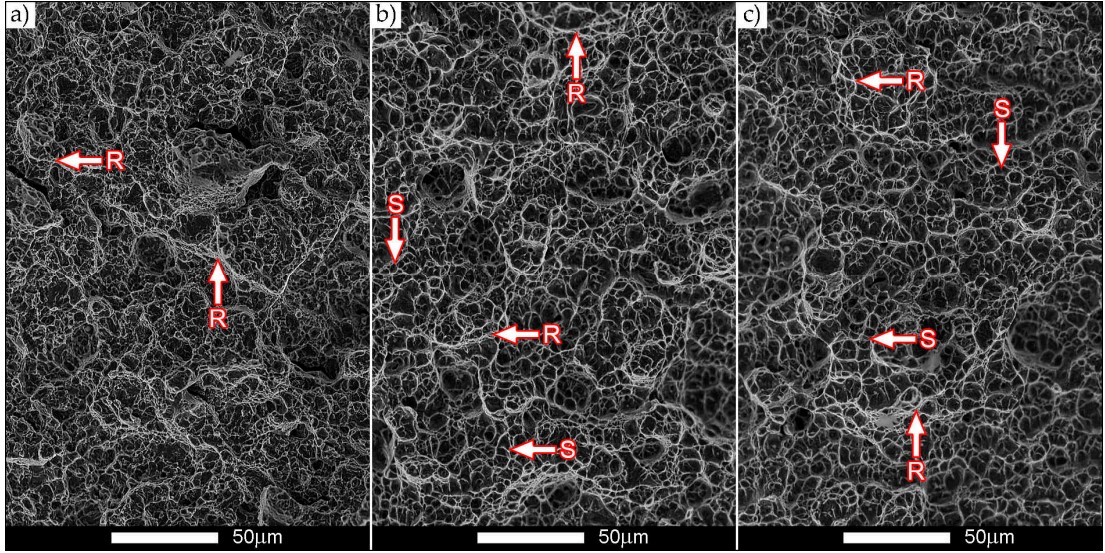

**Figure 27.** SEM images of fracture surfaces of Hardox Extreme welded joints after heat treatment, shown in Figure 24, after impact testing at +20 °C. (**a**) Area marked with the frame A2; (**b**) area marked with the frame B2; (**c**) area marked with the frame C2. R—"river" system; C—microcracks with fine steps; S—"scaly" steps. Scanning microscopy, unetched.

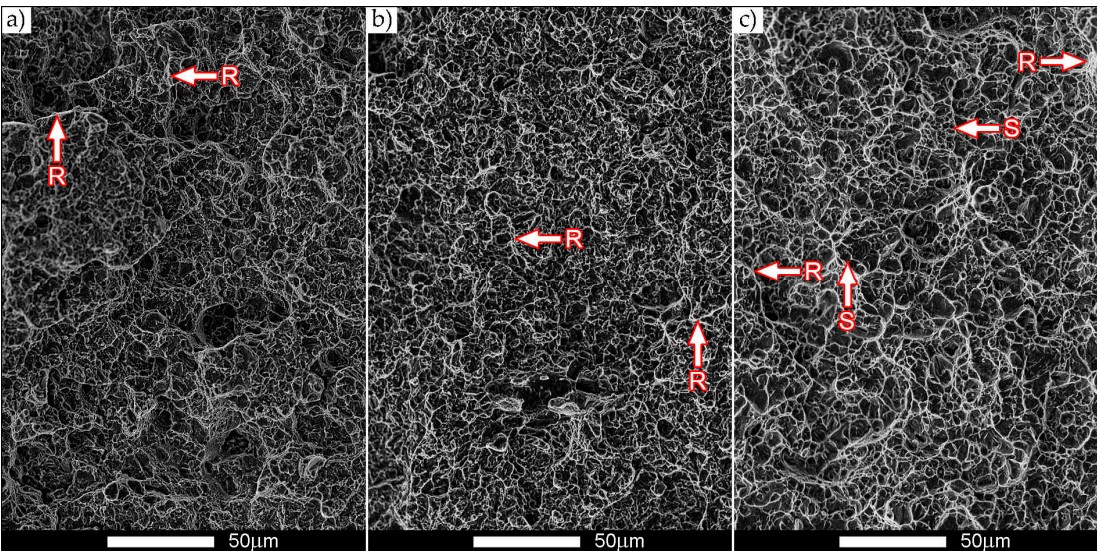

**Figure 28.** SEM images of fracture surfaces of Hardox Extreme welded joints after heat treatment, shown in Figure 24, after impact testing at −40 °C. (**a**) Area marked with the frame A4; (**b**) area marked with the frame B4; (**c**) area marked with the frame C4. R—"river" system; C—microcracks with fine steps; S—"scaly" steps. Scanning microscopy, unetched.

The described topography of a fracture surface is created by slips and decohesion, which results in appearance of microcracks in the planes {100} [29] and creation, after their separating walls are merged, of scales overlapping in a characteristic way. Parabolic contours of micro-voids indicate that the fracture is initiated by plastic deformation—slip—provoked by tangent forces in the process of fracture creation. It is worth mentioning that the cracking itself proceeds along the specific crystallographic planes.

In addition, it can be stated that identification of these planes is practically impossible because of the characteristic "river" relief occurring on fracture surfaces of the heat-treated specimens. This is caused by the fact that the meandering "river" system creates on a large area a system of micro-void coalescence characteristic for a plastic fracture. Mixed nature of the fractures contributes to the creation, during cracking, of steps increasing the amount of absorbed energy and thus decreasing the brittleness threshold.

## 4. Summary

It can be said on the grounds of the carried-out examinations that, in spite of limited weldability declared in the manufacturer's information materials, elements of the martensitic abrasive-wear resistant steel Hardox Extreme can be joined by welding techniques. It was shown in this elaboration that, through a proper selection of technology and welding parameters, it is possible to obtain an imperfection-free welded joint that is also characterized by very favorable strength indices. The above-mentioned assumptions were realized by proper selection of technology and welding parameters, as well as by use of additional heat treatment after welding, composed of normalization followed by volumetric quenching in oil and low tempering (stress-relieving). In the considered case, the applied thermal operations made it possible to restore, in the entire area of the welded joint, a structure and—to a large degree—a hardness level similar to those of the base material. The presented statement was positively verified in course of the carried-out strength and impact testing. The problems of execution and heat treatment of welded joints made of Hardox Extreme sheet metal can be characterized in the following ways:

- After welding, a highly morphologically diversified microstructure occurs in the entire area of the welded joint that shows, in comparison to the base material, lower hardness levels, tensile strength and impact energy. The obtained and relatively low mechanical properties of the welded joint make it possible to state that welding operations result in lowering the abrasive-wear resistance of the steel Hardox Extreme. The expected drop can occur in both the weld material and the area of base material directly adjacent to the very wide heat-affected zone.

- The additional thermal treatments carried-out after welding make it possible to favorably modify the structure in the entire welded joint and wide heat-affected zone, to obtain the structures similar to those of Hardox Extreme steel in the as-delivered condition from the manufacturer's plant.

- The obtained average hardness level of the welded joint after welding, amounting to only 17 J/cm$^2$ at +20 °C, clearly indicates susceptibility of this steel to brittle cracking. From practical point of view, it excludes application of welding techniques for joining Hardox Extreme sheets (irrespective of the relatively very high average tensile strength $R_m$ = 1278 MPa obtained) with no additional heat treatment operations. The above statement is additionally confirmed by the performed fractographic analysis.

- Examination results of heat-treated welded joints of Hardox Extreme steel indicate a possibility of restoring structural, mechanical, and impact properties "degraded" as a result of welding to the level corresponding to the base material. In the case of tensile strength, the obtained result $R_m$ = 1831 MPa makes a good reason for undertaking the problems of welding and heat treatment of the considered steel. An additional justification of this question is also obtained through heat treatment of other mechanical properties that are much better than those existing in the as-welded condition. In spite of a significant increase of the $R_m$ value (and also of the yield point, not cited in the reference), a nearly 12% increase percentage reduction of area reaching $Z$ = 29.1% (Table 5) was noted, as well as an increase of impact strength at ambient temperature to $KCV$ = 27 J/cm$^2$.

It is worth noting that the brittleness threshold of constructional materials is accepted as impact strength of 35 J/cm$^2$ [30], which results from maintaining at least 50% share of ductile fracture. Therefore, the obtained impact strength value after heat treatment and results of fractographic analysis make it possible to conclude that there was an occurrence of a favorable "shift" of plastic properties of the welded joint beyond the accepted brittleness threshold.

Irrespective of the conclusions formulated above, it should be also mentioned that the examination results of the steel Hardox Extreme, presented in this paper, constitute a fragment of the cycle of research works concerning low-alloyed martensitic, abrasive-wear resistant steels with an addition of boron, realized by the author for several years. Thus, it can be stated that problems of weldability, announced before, are not limited to the steel Hardox Extreme only, but concern almost the entire considered group of materials. The materials already considered by the author include the steels: Hardox 400, Hardox 500, Hardox 600, HTK700H, HTK900H, AR400, XAR 600, Creusabro 4800, Creusabro 8000, TBL Plus, B27, Brinar 400, Brinar 500, and others. In almost all of the considered cases, problems with welding of these steels were observed, although the manufacturers declared their weldability. Therefore, attempts to weld high-strength abrasive-wear resistant steels with subsequent post-weld heat treatment seem to be very well grounded. In addition, it is also worth considering the application of advanced methods of hybrid welding and dedicated induction heating stations for welding and heat treatment of the above-mentioned materials. Optimum usage of this type infrastructure would make it possible to apply welding techniques for joining high-strength abrasive-wear resistant martensitic steels, while maintaining their very profitable mechanical properties and usable features.

**Funding:** This research received no external funding.

**Acknowledgments:** The author would like to thank Józef Ptak from Stal-Hurt S.C. company for providing sheets of Hardox steels and for Zbigniew Konat and Eugeniusz Szymanowicz from the Wroclaw Shipyard for assistance in the implementation of welded joints.

**Conflicts of Interest:** The author declares no conflict of interest.

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
