# Peer review of "Structural Aspects of Execution and Thermal Treatment of Welded Joints of Hardox Extreme Steel"

_metals, doi:10.3390/met9090915_

Round 1
Reviewer 1 Report
Please, see attached file

Author Response
Dear Reviewer,
Please, see attached file.
With best regards,
Łukasz Konat

Reviewer 2 Report
The manuscript is generally well written, the equipment and methodology are clearly described and the discussions are sustained by the obtained results and by references from the scientific literature.
I recommend publishing this manuscript after some minor corrections on spelling and text formatting. Also, some of the instructions for authors were left at the end of the manuscript.
Author Response
Dear Reviewer,
I would like to thank you very much for reviewing my manuscript and at the same time I apologize for any editorial errors in the form of improperly formatted tables, fonts etc. In the final version of the manuscript I tried to remove all these artifacts.
Again thank you very much.
With best regards,
Łukasz Konat
Reviewer 3 Report
Generally, the data provided by the manufacturer is reliable. Why are there significant differences in the amounts of chemical components in Tables 2 and 3.
Heat treatment: There is no significant difference in compositions of Hardox 600 and Extreme. The heat treatment applied in this study is not special, but normal. It seems that lots of study on Hardoc has been performed. Compared to studies on other Hardox steels, 500, 600, please describe what original contents of this study are.
Explain as schematic diagrams how tensile and impact specimens were produced from the welded plate.
Line 118: … all the thermal operations were carried out in gas tight …. Of 99.95% argon.
Table 5: Normalization 800℃/1h/air, and tempering 100℃/20h/Airè The air means argon?
Figure 8: 9, 14, 15, 18, 19, 20: Photos in 50mm are not enough to reveal detail microstructures, explain in more detail how the microstructures were identified as bainite, troostite, bainite, martensite.
Page 18 : indicate “river” microcracks, slides in Figure 23.
The author said that he is currently realizing examinations of Hardox welded joints with regard to their abrasive wear in real conditions of soil abrasive mass. In my opinion, the author should add some results for wear in order for this paper to be published.
Author Response
Dear Reviewer,
Please, find attached file.
With best regards,
Łukasz Konat

Reviewer 4 Report
1. It was not shown from what welded joint place the samples for the Impact test were cut. for example, was the V-notch of specimens in the [0;0] zone of weld joint (fig.2) or in other place?
2. Fig.4, is not in correct scale for presented content, because it not correlate with fig.3.: not all zones, presented in fig.3 are visible in fig.4. It is important in presented case for to compare the differences which brings the thermal treatment.
3. Is no discussion about the results: what factors determine such changes of structure and properties. For example, was it influence of initial structure, or post treatment just after second seam welding on the opposite side of specimen's, or specific conditions of thermal treatment. By the way, what was tempering conditions? It not named in the article.
Why you use two cutting methods for specimens preparation: HEAWS and EE? Did the cutting method impact the some results?
Author Response

(The authors gave the same response as above.)

Round 2
Reviewer 3 Report
In Fig. 2
- Please mark the rolling direction of the plate.
- Please mark the dimension of the width of weld bead.
- Please mark the shape of the impact and tensile specimens.
Author Response
Dear Reviewer,
As suggested, I placed all markers in Fig. 2, except for the direction of sheet rolling. Hardox steels are rolled in both directions and without additional testing, it is practically impossible to indicate the main direction of thermomechanical treatment.
With best regards,
Łukasz Konat